# Flame Retardant Functionalization of Microcrystalline Cellulose by Phosphorylation Reaction with Phytic Acid

**DOI:** 10.3390/ijms22179631

**Published:** 2021-09-06

**Authors:** Hua-Bin Yuan, Ren-Cheng Tang, Cheng-Bing Yu

**Affiliations:** 1College of Textile and Clothing Engineering, Soochow University, Suzhou 215021, China; huabinyuan19@163.com; 2China National Textile and Apparel Council Key Laboratory of Natural Dyes, Soochow University, Suzhou 215123, China; 3Jiangsu Engineering Research Center of Textile Dyeing and Printing for Energy Conservation, Discharge, Reduction and Cleaner Production (ERC), Soochow University, Suzhou 215123, China; 4School of Materials Science and Engineering, Shanghai University, Shanghai 200444, China

**Keywords:** microcrystalline cellulose, phytic acid, phosphorylation, flame retardant, functionalization

## Abstract

The functionalization of microcrystalline cellulose (MCC) is an important strategy for broadening its application fields. In the present work, MCC was functionalized by phosphorylation reaction with phytic acid (PA) for enhanced flame retardancy. The conditions of phosphorylation reaction including PA concentration, MCC/PA weight ratio and temperature were discussed, and the thermal degradation, heat release and char-forming properties of the resulting PA modified MCC were studied by thermogravimetric analysis and pyrolysis combustion flow calorimetry. The PA modified MCC, which was prepared at 90 °C, 50%PA and 1:3 weight ratio of MCC to PA, exhibited early thermal dehydration with rapid char formation as well as low heat release capability. This work suggests a novel strategy for the phosphorylation of cellulose using PA and reveals that the PA phosphorylated MCC can act as a promising flame retardant material.

## 1. Introduction

Agricultural and forestry wastes are a kind of low-cost, widely sourced and recyclable bio-based materials [1], and usually include straw, rice straw, waste wood, wheat bran, rice husk, bagasse, etc. [2]. These wastes are mainly composed of cellulose, hemicellulose and lignin [3], which are an abundant and cheap raw material for the preparation of microcrystalline cellulose (MCC) [4]. Due to its abundant sources, odorlessness, renewability, excellent mechanical properties and high reactivity, MCC has been widely used in food, cosmetic, pharmaceutical, beverage, packaging, chemical, and polymer composite industries [5]. In order to make MCC suitable for these uses in these fields, various chemical modifications are often required [6,7]. Among them, the flame retardant (FR) functionalization yielding FR MCC is one of the important modifications.

FR MCC can be used as an additive of polymers and composite materials for their enhanced FR performance [8]. Compared with traditional toxic and harmful halogen-containing FR agents, FR MCC additives have the advantages of being biodegradable, non-toxic and harmless. Therefore, the preparation and application of FR MCC have received extensive attention. For example, acrylated MCC functionalized by 9,10-dihydro-9-oxa-10-phosphaphenanthrene-10-oxide could be coated on paper and wood, protecting them from fire [9]. The finishing using the mixture of phosphorylated nano-cellulose and chitosan prevented thermal degradation of jute fabric due to the formation of a superficial char layer [10]. Phosphorylated nano-cellulose was incorporated into polyurethane foam, which worked as reinforcing and FR agents [8]. MCC was modified by 3-aminopropyltriethoxy silane coupling agent and ammonium polyphosphate, and the as-prepared product acted as FR and reinforcing agents for epoxy resin [11]. If MCC grafted with methacrylic acid, together with ammonium polyphosphate, was blended into polylactic acid by melt compounding, the resulting composite had a good FR performance [12].

Phytic acid (PA), also known as inositol hexakisphosphoric acid, is widely found in various beans and grains [13]. PA is a cyclic substance with six phosphate groups connected to a cyclohexane ring. Due to its extremely high phosphorus content [13], biological nature and biocompatibility, PA is undoubtedly an environmentally benign FR agent [14]. The application of PA alone or in combination with other agents for the FR improvement of cellulose materials has been reported [15,16,17]. When PA was used as a FR agent for cotton, it was able to initiate the charring of cotton at a decreased temperature of the thermal degradation and increase the char yield of cotton, leading to a significant improvement in the flame retardance of cotton [15,16]; however, the PA treated cotton had poor resistance to washing [15]. Recent reports showed that the combination of PA with chitosan [18,19], egg white protein [20], biochar [15] and silane sol [21] endowed cellulosic textiles with excellent FR properties. In addition, the mixtures of PA with hydrolyzed collagen [22], melamine [23] and polyaniline [24] have been proved to be very effective in retarding the combustion of wood, bamboo slice and paper, respectively.

An alternative approach to prepare FR cellulose materials is the phosphorylation of cellulose [25], which can be readily achieved by using phosphorylating agents such as phosphoric acid, phosphoric acid, sodium trimetaphosphate and phosphate ester at high temperatures [25,26,27,28]. In this work, MCC was phosphorylated with PA in order to obtain a PA modified MCC (PA-MCC) product with good char-forming ability and FR properties.

## 2. Results and Discussion

Figure 1 illustrates the MCC and PA structures as well as the schematic diagram of PA-MCC preparation. This study mainly explored the conditions of phosphorylation reaction as well as the thermal stability and char formation properties of PA-MCC. The functional groups of PA-MCC were characterized by Fourier transform infrared (FT-IR) spectroscopy, and the P content of PA-MCC was determined by the inductively coupled plasma mass spectrometry (ICP-MS). The thermal stability, char formation and heat release properties of PA-MCC were studied by thermogravimetric (TG) analysis and pyrolysis combustion flow calorimetry (PCFC).

### 2.1. PA Concentration

Figure 2a shows the FT-IR spectra of MCC and PA-MCC. PA-MCC showed the C=O stretching vibration at 1715 cm^−1^, which is likely due to the formation of carbamate groups caused by the reaction of cellulose and urea [29,30], and the partial oxidation of cellulose at high temperatures [31,32]. A shoulder peak appeared at 995 cm^−1^ for PA-MCC while it was not observed for MCC, corresponding to the P-OH bond of phosphate groups in PA reacted with MCC [30]. However, the signals of P=O and P-O-C bonds at 1250–1000 cm^−1^ [30] could not be clearly distinguished. Because the intensity of P=O and P-O-C bonds is related to the P content of phosphorylated cellulose [32,33], their peaks in the present study did not appear due to the low P content and the overlapping effect of C-O-C groups in cellulose. The changes of the FT-IR spectra of PA-MCC confirm the phosphorylation reaction of MCC and PA as well as the side reactions. The phosphorylation reaction can be further demonstrated by the analysis of the P content in the Section 2.3. Reaction Temperature. When PA concentration was 70%, MCC underwent hydrolysis in PA solution at high temperatures, resulting in a very small amount of remained solid products. Therefore, the maximal concentration of PA was limited to 50%.

Figure 2b,c illustrate the thermal performance of PA-MCC products characterized by TG in nitrogen and air. In the process of heating, the weight loss caused by water evaporation occurred below 150 °C. The main weight loss of MCC took place between 310 and 370 °C in nitrogen, and between 300 and 360 °C in air, respectively, which is due to the pyrolysis of cellulose molecules. Regardless of in nitrogen or in air, PA-MCC samples showed lower onset and rapid degradation temperatures than MCC, which is attributed to the fact that PA has poorer thermal stability than cellulose and decomposes firstly to produce phosphoric acid [34], promoting the pyrolysis of MCC. It is worth noting that PA-MCC displayed higher residual weight than MCC due to the PA-induced carbonization of MCC. Furthermore, as the concentration of PA increased, more PA molecules were bound to MCC. Such PA-MCC produced more phosphoric acid in the process of thermal degradation, leading to the increasing dehydration and carbonization of MCC. Therefore, it is concluded that PA-MCC has good char-forming ability in the thermal pyrolysis, which can hinder the heat and oxygen transfer, and prevent the formation of volatiles, so that an improvement in the flame retardant characteristics of phosphorylated MCC can be expected.

The thermal combustion properties of PA-MCC were tested by PCFC. The results were shown in Figure 2d and Table 1. Compared with MCC, PA-MCC exhibited significant reductions in heat release capacity (HRC), peak heat release rate (pHRR), and total heat release (THR); and furthermore, these parameters displayed a great decrease with increasing PA concentration. It means that PA modification can effectively control the heat release intensity of MCC, thereby significantly reducing the risk of ignition. In addition, the amount of char residue also increased with increasing PA concentration. It is not difficult to understand that the greater the amount of char, the less the part of combustible volatiles. Therefore, the excellent char-forming ability of PA-MCC is beneficial to the improvement of FR performance.

### 2.2. Dosage Ratios of MCC to PA

Figure 3 shows the FT-IR, TG, and PCFC results of PA-MCC samples obtained at various weight ratios of MCC to PA. The highest intensity of the C=O stretching vibration at 1715 cm^−1^ was observed for PA-MCC obtained at a 1:3 weight ratio of MCC to PA (Figure 3a). However, the differences in the intensities of P-OH, P-O-C and P=O bands were difficultly distinguished for PA-MCC samples obtained at various weight ratios of MCC to PA.

Figure 3b,c illustrate the TG curve of PA-MCC samples. Regardless of in nitrogen or in air, PA-MCC samples displayed a decrease in onset temperature and an increase in weight loss with increasing PA proportion. Compared with the degradation in nitrogen, different PA-MCC samples exhibited a greater difference in residual weight at high temperatures. The residual weight increased firstly with increasing PA proportion, and then decreased with a further increase in PA proportion. At a 1:3 weight ratio of MCC to PA, the residual weight of PA-MCC was highest; for instance, the value at 500 °C was 27.3% while those were 11.4%, 24.0% and 17.6%, respectively at 1:2, 1:5 and 1:10 weight ratios of MCC to PA. This observation proves that PA-MCC prepared at a 1:3 weight ratio of MCC to PA exhibits the best char-forming ability.

The flammability characteristics characterized by PCFC are shown in Figure 3d and Table 2. With increasing PA proportion, the HRC, pHRR and THR of PA-MCC decreased firstly and then increased slightly; they were the minimum at a 1:3 weight ratio of MCC to PA. In contrast with HRC, pHRR and THR, the residual char yield showed an opposite trend, and it was highest for PA-MCC prepared at a 1:3 weight ratio of MCC to PA. The PCFC results had a good correlation with the TG analyses in air. Compared with MCC, all PA-MCC samples showed a significant reduction in HRC, pHRR and THR and an obvious increase in char yield. Therefore, the conclusion can be drawn that PA-MCC is a good FR material, and it possesses low combustibility and fire hazard.

### 2.3. Reaction Temperature

Figure 4 shows the FT-IR, TG, and PCFC results of PA-MCC samples prepared at various reaction temperatures. PA-MCC samples prepared at 90–96 °C (Figure 4a) showed the highest intensity of the C=O stretching vibration at 1715 cm^−1^. However, the differences in the absorption intensities of P-OH, P-O-C and P=O groups still could not be distinguished for PA-MCC samples prepared at various temperatures.

In order to confirm the reaction between PA and MCC, the P content of PA-MCC was determined by ICP-MS. Figure 5 shows that PA-MCC prepared at 90–96 °C had obviously higher P content than that prepared at 85 °C. It implies that the phosphorylation reaction between PA and MCC occurs, and it is very sensitive to temperature. When the reaction occurred at 90 °C, the highest P content was achieved, indicating that the optimal reaction temperature is 90 °C. In previous reports, the phosphorylation of wheat starch with phytate in aqueous suspension [35], the phosphorylation of waxy starch with PA by dry heat treatment at 120 °C for 24 h [36], and the phosphorylation of maize, rice, and potato starches with phytate by dry heat treatment at 130 °C for 12 h [37] were studied, and the P content of the products was 0.04–0.05%, 0.0093%, and 0.067–0.089%, respectively. Compared with the phosphorylation of starches with PA and phytate, the PA-phosphorylated cellulose in the present work had a high P content.

In the TG curves (Figure 4b,c), the PA-MCC obtained at 90 °C showed the best thermal stability above 325 °C and the highest residual weight. The heat release ability of PA-MCC determined by PCFC is shown in Figure 4d and Table 3.

In sharp contrast with PA-MCC obtained at 85 °C, PA-MCC samples obtained at 90–96 °C showed a significant reduction in HRC, pHRR and THR. The sample prepared at 90 °C yielded the lowest THR and the highest residual char yield. Both the TG and PCFC analyses reveal that 90 °C is the optimal temperature for the reaction of PA and MCC. The conclusion is in good agreement with the analysis of the P content.

## 3. Materials and Methods

### 3.1. Materials

PA (70%) and MCC (50 μm) used in the experiment were purchased from Chengdu Ai Keda Chemical Technology Co. Ltd., Chengdu, Sichuan Province, China and Shanghai Meryer Chemical Technology Co. Ltd., Shanghai, China, respectively. Dicyandiamide was purchased from Shanghai Ling Feng Chemical Reagent Co. Ltd., Shanghai, China; urea, sodium hydroxide and cyclohexane were bought by Jiangsu Qiang Sheng Chemical Co. Ltd., Changshu, Jiangsu Province, China; the above reagents were all analytically pure.

### 3.2. FR Functionalization of MCC Using PA

Prior to the FR functionalization of MCC by phosphorylation reaction with PA, the swelling pretreatment by alkali was performed. A certain amount of MCC was placed in a 14% sodium hydroxide solution and stirred continuously until it was evenly dispersed; then, the alkaline MCC solution was sealed and stood at room temperature for 24 h. The MCC was diluted with an appropriate amount of deionized water and then washed several times until it was neutral. Later, a wet MCC cake was obtained by filtration. After drying at 70 °C for 72 h in the oven and grinding, the pretreated MCC powder was obtained.

70% PA solution was accurately weighed and added into a three-necked flask which was placed in a glycerol bath. Afterward, pretreated MCC powder, catalysts (mixture of dicyandiamide and urea) and water-carrying agent (cyclohexane) were successively added into the three-necked flask. The above mixture was stirred in a magnetic stirrer at the desired temperature for 4 h. During the reaction, cyclohexane floated on the mixture of PA and MCC. Then the temperature was lowered to 72 °C, an appropriate amount of deionized water was added into the three-necked flask, and the stirring was continued at this temperature until the reaction product was uniformly dispersed in the water.

The reaction product mentioned above was filtered, and the separated filter cake was thoroughly washed with deionized water to completely remove cyclohexane and unreacted PA. The resulting filter cake was dried at 70 °C for 72 h in the oven and ground. Thus, a PA modified MCC solid powder (PA-MCC) was obtained.

In order to obtain the optimized reaction conditions, three PA concentrations (70%, 50% and 30%), four dosage ratios of MCC to PA (1:2, 1:3, 1:5 and 1:10), and four temperatures of reaction media (85, 90, 94 and 96 °C) were discussed.

### 3.3. Measurements

Pyrolysis combustion flow calorimetry (PCFC, FTT0001, Fire Testing Technology Ltd., East Grinstead, UK) was used to study the heat release properties of PA-MCC according to ASTM D7309. Each sample was taken for three measurements and the average value was taken. Thermogravimetric (TG) analysis (Diamond 5700, Perkin-Elmer Inc., Waltham, MA, USA) was used to study the thermal performance of PA-MCC in nitrogen and air atmospheres from 30 to 600 °C with a heating rate of 10 °C per minute. Fourier transform infrared spectroscopy (FT-IR, Nicolet 5700, Thermo Fisher Scientific Inc., Waltham, MA, USA) was used to analyze the chemical structures of PA-MCC ranging in 500–4000 cm^−1^. Inductively coupled plasma mass spectrometry (ICP-MS 7700, Agilent Technologies Inc., Santa Clara, CA, USA) was used to determine the P content of PA-MCC, and the measurement was performed at Shiyanjia Lab (Hangzhou, China).

## 4. Conclusions

In this study, the conditions of the phosphorylation of MCC with PA were explored, and the thermal stability and pyrolysis combustion properties of the resulting PA-MCC products. FT-IR spectroscopy and P content analysis confirmed the reaction between PA and MCC. TG analysis showed that the thermal stability of PA modified MCC at high temperatures was significantly improved. PCFC test proved that the heat release capacity of PA-MCC was far below that of MCC. By combining the FI-IR, P content, PCFC and TG analyses, the optimal conditions for the reaction of MCC and PA were obtained as follows: PA concentration, 50%; MCC/PA weight ratio, 1:3; temperature, 90 °C. The PA-MCC prepared in the optimal conditions, which had a P content of 0.63%, showed good char-forming ability and low heat release performance during the thermal degradation. This study reveals that PA-MCC is a good FR material, and it possesses low combustibility and fire hazard. In addition, this study proposes a novel approach to prepare the FR MCC material by the phosphorylation reaction using PA.

## Figures and Tables

**Figure 1 ijms-22-09631-f001:**
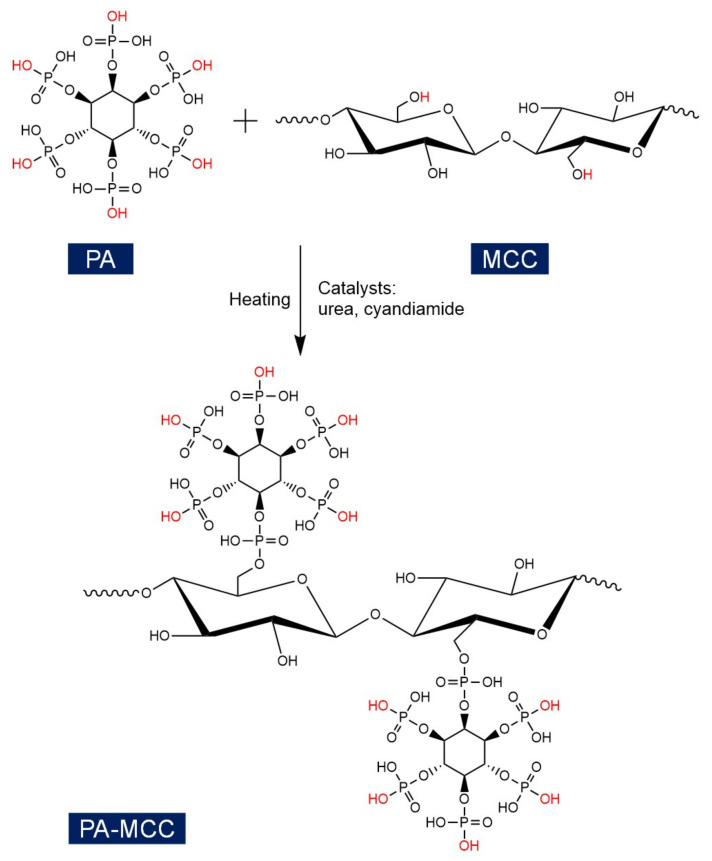
Schematic diagram of the preparation of PA modified MCC.

**Figure 2 ijms-22-09631-f002:**
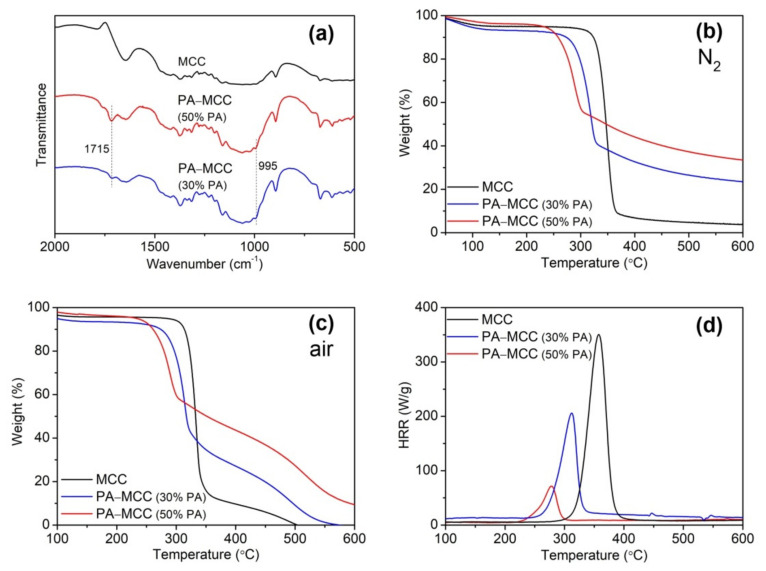
FT-IR (**a**), TG (**b**,**c**), and PCFC (**d**) analyses of PA-MCC obtained at various concentrations of PA. (PA concentration: 30% and 50%; MCC/PA weight ratio: 1:3; PA/urea/dicyandiamide weight ratio: 15:1:2; 90 °C; 4 h).

**Figure 3 ijms-22-09631-f003:**
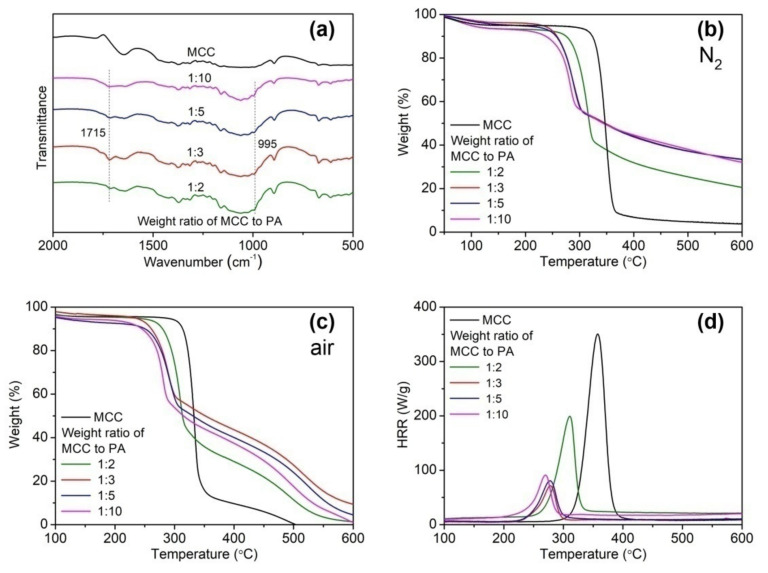
FT-IR (**a**), TG (**b**,**c**), and PCFC (**d**) analyses of PA-MCC obtained at various weight ratios of PA to MCC. (PA concentration: 50%; MCC/PA weight ratio: 1:2, 1:3, 1:5, and 1:10; PA/urea/dicyandiamide weight ratio: 15:1:2; 90 °C; 4 h).

**Figure 4 ijms-22-09631-f004:**
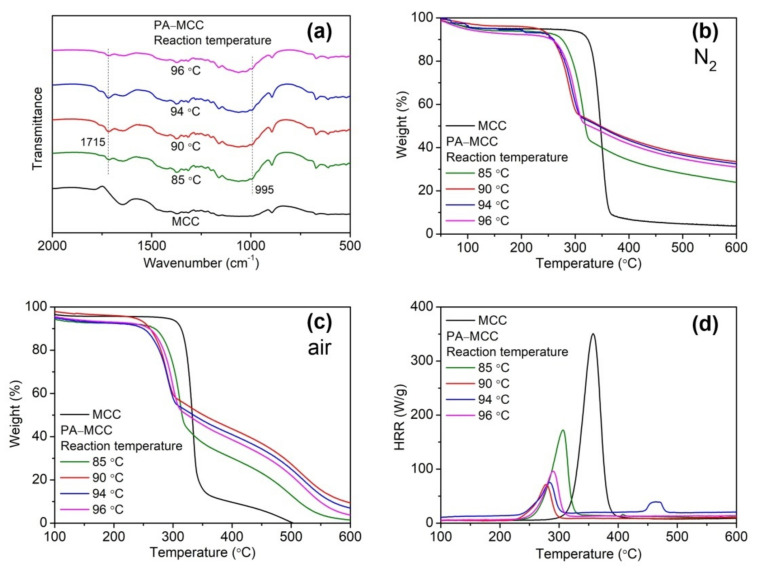
FT-IR (**a**), TG (**b**,**c**), and PCFC (**d**) analyses of PA-MCC obtained at various reaction temperatures. (PA concentration: 50%; MCC/PA weight ratio: 1:3; PA/urea/dicyandiamide weight ratio: 15:1:2; reaction temperature: 85, 90, 94 and 96 °C; 4 h).

**Figure 5 ijms-22-09631-f005:**
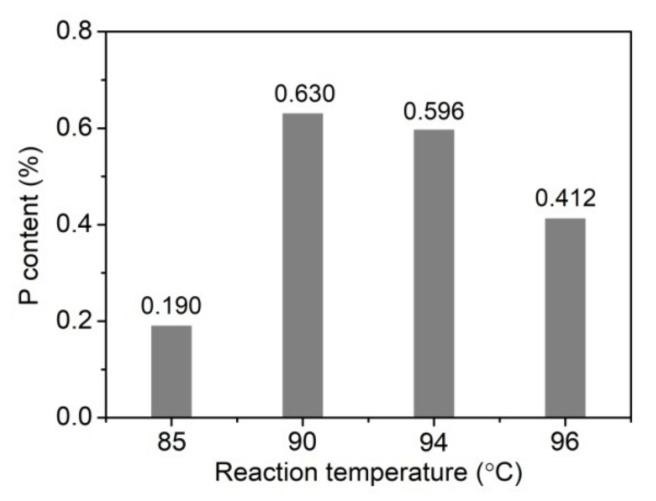
P content of PA-MCC obtained at various reaction temperatures.

**Table 1 ijms-22-09631-t001:** PCFC data of PA-MCC obtained at various concentrations of PA.

Sample	HRC (J/(gK))	pHRR (W/g)	THR (kJ/g)	Char Residue (%)
MCC	361.0	343.3	12.7	1.1
PA-MCC (30% PA)	197.0	193.5	7.6	17.5
PA-MCC (50% PA)	68.0	64.7	2.0	33.5

**Table 2 ijms-22-09631-t002:** PCFC data of PA-MCC obtained at various weight ratios of PA to MCC.

Sample	HRC (J/(gK))	pHRR (W/g)	THR (kJ/g)	Char Residue (%)
MCC	361.0	343.3	12.7	1.1
PA-MCC (2:1)	186.0	184.1	7.3	20.5
PA-MCC (3:1)	68.0	64.7	2.0	33.5
PA-MCC (5:1)	77.0	72.7	2.5	31.6
PA-MCC (10:1)	80.0	77.9	3.0	31.9

**Table 3 ijms-22-09631-t003:** PCFC data of PA-MCC obtained at different reaction temperatures.

Sample	HRC (J/(gK))	pHRR (W/g)	THR (kJ/g)	Char Residue (%)
MCC	361.0	343.3	12.7	1.1
PA-MCC (85 °C)	173.0	164.6	6.0	22.5
PA-MCC (90 °C)	68.0	64.7	2.0	33.5
PA-MCC (94 °C)	63.0	60.6	3.1	32.8
PA-MCC (96 °C)	90.0	87.2	3.0	31.0

## Data Availability

The data are available upon request.

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
