# Peer review of "Flame Retardant Functionalization of Microcrystalline Cellulose by Phosphorylation Reaction with Phytic Acid"

_ijms, 2021, doi:10.3390/ijms22179631_

Round 1
Reviewer 1 Report
This article is interesting. However, it contains many shortcomings that need to be corrected.
Introduction, second paragraph. “FR MCC additives have… low cost”. Remark: This statement is not correct because production cost even of unmodified MCC is enough high, $5-8 per kg. After phosphorylation, the cost of the final product can double, at least. Thus, FR MCC is an expensive additive, and therefore the statement on its “low cost” should be removed.
Introduction, the last sentence on p.1. “For example, acrylated MCC functionalized by … could be coated on paper and wood protecting them fire; Remark: This sentence contains grammatical errors and should be corrected, as follows: “For example, MCC functionalized by … can coat paper and wood protecting them from fire
Line 46. Remark: Put a point after [9]. Start the next sentence with a capital letter.
Line 48. Remark: Put a point after [10]. Start the next sentence with a capital letter.
Line 49. Remark: Put a point after [8]. In addition, replace the word ”reinforce” with the word “reinforcing”.
Line 51. Remark: The phrase contains grammatical errors and should be corrected, as follows: “which were introduced into epoxy resin and acted as FR and reinforcing agents”. In addition, put a point after [11] and start the next sentence with a capital letter.
Lines 51-53. The sentence presented after [11] is unclear and should be corrected, as follows: “If MCC modified with ammonium polyphosphate and grafted with methacrylic acid is blended with polylactic acid, the resulting composite has a good FR performance”.
Line 54. Remark: The preferred English synonym of PA is “inositol hexaphosphate acid”. Use this synonym instead of “inositol hexakisphosphoric acid”.
Line 55-56. Remark: The description of molecular structure of PA has some errors and should be corrected, as follows: “ PA is a cyclic substance carbohydrate with six phosphate groups symmetrically connected to a cyclohexane ring. in its molecular structure
Line 57… biological source nature… Remark: Remove the unnecessary word “source”.
Line 61…. charring of cotton at a low temperature during the thermal degradation… Remark: This phrase should be improved, as follows: …. “charring of cotton at a decreased temperature of the thermal degradation”…
Line 64. Remark: Use “…the combination of PA”… instead of …“the combined application of PA”…
Lines 74-75. Figure 1 illustrates the MCC and PA structures as well as the schematic diagram of PA‐MCC preparation. Remark: Since illustrations relate to Results, this sentence should be moved to the section “Results and Discussion”, before Fig.2
Line 84. C=O stretching vibration at 1715 cm‐1 which is attributable to the carbamate groups. Remark: Yes, it is known that IR peak at 1715 cm‐1 is related to carbonyl groups. However, it has not been proven that they are carbonyls of the carbamate groups of MCC. An additional argument is that the carbamate groups are intermediate and should disappear as a result of phosphorylation of MCC, since there are no carbamate groups in Fig. 1. Thus, I recommend give the general interpretation of IR peak at 1715 cm‐1 as vibrations of C=O groups in FA-MCC.
Lines 90-91. As PA concentration… Remark: Replace “As” with “When”, i.e. When PA concentration was 70%, MCC underwent hydrolysis in PA solution at high temperatures, resulting in a very small amount of remained solid products.
Lines 92-93. So, the relative characterizations were not conducted. Remark: Remove this sentence as unnecessary and replace it with a new sentence “Therefore, the maximal concentration of PA was limited to 50%.
Line 107 …decomposes firstly at high temperatures to produce phosphoric acid... Remark: Remove “at high temperatures” as unnecessary, i.e. write …”decomposes firstly to produce phosphoric acid”…
Lines 113-115. …which can play the role of hindering heat transfer, isolating oxygen, and preventing volatiles formation, so as to be expected to improve the thermal and flame retardant performances. Remark: This phrase should be re-edited, as follows: …”which can hinder the heat and oxygen transfer, and prevent the formation of volatiles, so that an improvement in the flame retardant characteristics of phosphorylated MCC can be expected.
Line 121 …thereby significantly reducing the combustion hazard. Remark: This phrase should be edited, as follows: …”…thereby significantly reducing the risk of ignition”.
Lines 123-124 …the greater the amount of char, the less the combustible part. Remark: Edit this phrase, as follows: …”the greater the amount of char, the less the part of combustible volatiles”.
Author Response
Dear Editors, and Reviewers,
Thank you very much for your letter and the reviewers’ comments on our manuscript ijms-1340759. Taking the reviewers’ comments into consideration, we have supplemented the analysis of P content, and made modification on the original manuscript. We hope the revised manuscript can meet the requirements for approval. In the revised manuscript, the revised portion is marked in red for easy reviewing purpose.
The responses to the reviewer’s comments can be found in the attached word file.
Should you have any questions, please contact us without hesitation.
Once again, we appreciate very much for your time in editing our manuscript, and thank the reviewers for the valuable suggestions and comments. I am looking forward to hearing from your final decision when it is made.
Sincerely yours,
Ren-Cheng Tang, August 23, 2021

Reviewer 2 Report
This manuscript reports on various conditions of phosphorylation of microcrystalline cellulose (MCC) by phytic acid. The phosphorylated MCC was studied by TGA and PCFC only. I recommend rejecting this paper because of serious flaws in phosphorylation procedure and poor characterization of the phosphorylated MCC.
The phosphorylation is performed in water added with cyclohexane. Water boils at 100 C, cyclohexane boils at 81 C. How is it possible to perform phosphorylation at 120-150 C?
After phosphorylation the cellulose was washed with water to remove unreacted phytic acid. How much acid was removed and how much of it reacted with cellulose?
Urea and dicyandiamide were used as catalysts. Figure 1 doesn't show any reaction of cellulose with these chemicals. In contrast, C=O stretching of FTIR is attributed to urea.
Strangely, FTIR doesn't show any peaks attributed to P=O and P-O-C despite the fact that these are very strongly absorbing groups. It seems that FTIR is not useful tool for this study. In the the absence of phosphorus chemical analysis it is impossible to say what do tested MCC-PA represent.
Author Response

(The authors gave the same response as above.)

Round 2
Reviewer 2 Report
Accept as corrected